# Working dogs in dynamic on-duty environments: The impact of dark adaptation, strobe lighting and acoustic distraction on task performance

**Elizabeth L. Sheldon**[1‡]*, **Carla J. Hart**[1‡], **Anna Wilkinson**[1], **Carl Soulsbury**[1], **Victoria F. Ratcliffe**[2], **Daniel S. Mills**[1]

**1** School of Life and Environmental Sciences, University of Lincoln, Lincoln, United Kingdom, **2** Defence Science and Technology Laboratory, Salisbury, Wiltshire, United Kingdom

‡ ELS and CJH are contributed equally as first authors on this work.
* elizabeth-louise.sheldon@students.mq.edu.au

**Data Availability Statement:** The data is available at a public repository: Sheldon, Elizabeth (2022):

## Abstract

Sudden changes in sound and light (e.g., sirens and flashing police beacons) are a common component of working dogs' on-duty environment. Yet, how such stimuli impact dogs' ability to perform physical and cognitive tasks has not been explored. To address this shortcoming, we compared the accuracy and time taken for twelve dogs to complete a complex physical and cognitive task, before, during and after exposure to three 'real-world' stimuli: an acoustic distractor (85dB), white strobe lighting (5, 10 & 15 Hz), and exposure to a dazzling white, red, or blue lights. We found that strobe lighting, and to a greater extent, acoustic distraction, significantly reduced dogs' physical performance. Acoustic distraction also tended to impair dogs' cognitive performance. Dazzling lights had no effect on task performance. Most (nine out of twelve) dogs sensitised to the acoustic distraction to the extent of non-participation in the rewarded task. Our results suggest that without effective distractor response training, sudden changes in noise and flickering lights are likely to impede cognitive and physical task performance in working dogs. Repeated uncontrolled exposure may also amplify these effects.

## Introduction

Working dogs are often required to perform trained tasks in dynamic environments, with little time to acclimatise to their surroundings. Changes to the working environment could be distracting, stressful, or affect sensory integration in dogs, which may reduce their ability to perform complex physical and cognitive tasks. Changes to the weather [1, 2], the dog's handler [3], non-target scents [4, 5] and distractor items [6] have been shown to impact task performance in working dogs. Yet, to date, the performance effects of changes in two salient, and near ubiquitous stimuli; background noise and light levels, have not been assessed. Because dogs have been selectively bred over thousands of years to work cooperatively with humans [7], there is tendency to assume that the stimuli that distract/disorient us are the same as those

Working dogs in dynamic on-duty environments. figshare. Dataset. https://doi.org/10.6084/m9. figshare.21717776.v1.

**Funding:** Funding was provided by UK Ministry of Defence (MOD) who plated a role in study design and preparation of the manuscript.

**Competing interests:** The authors have declared that no competing interests exist.

that will impair task performance in dogs. Yet, many functional differences in the visual and auditory systems of humans and dogs exist [8, 9]. To maximise working dogs' performance in dynamic and unpredictable working environments, it is necessary to gain a solid understanding of how 'real-world' changes in light and sound impact the accuracy, efficiency, and repeatability of dogs' task performance.

Compared to humans, dogs can see better in low light conditions, yet they take longer to recover in dim lighting following 'photo bleaching' [10]. Photo bleaching occurs when photopigment temporarily changes its form and loses its functionality following exposure to bright light [11]. Numerous ocular properties of dogs (e.g., the presence of a tapetum, a larger pupil and more rod dominated retina compared to humans [10]) favour performance in low lights, but also potentially contribute to an extended visual recovery time (although dogs might have biochemical differences that affect rhodopsin regeneration time [12]). Indeed, while the recovery of dark-adapted vision may take over an hour for dogs [13], it takes ~ 30 min in humans [14]. Consequently, it might bright light interruptions within low light conditions may affect dogs more than humans.

Working dogs are often brought from bright environments directly into darker working conditions (e.g., cave rescue dogs [15]), and are also exposed to 'dazzling' lights during night shifts (e.g., exposure to torches, car headlights or emergency service lights). It is therefore important to establish whether sudden changes in light have a functional effect on dogs' task performance. 'Dark adaptation' (the process of recovery of dark vision following photobleaching) may be associated with impaired performance in visual and non-visual-tasks for dogs (e.g., those requiring motor skills [16]) due to its effects on visual acuity and/or by being generally distracting [17]. However, to our knowledge, the impact of dark adaptation on task performance has never been tested in dogs. Further, it is unclear whether exposure to light colours (e.g., the red, blue or white emergency lights encountered by police dogs) differentially impact dogs' task performance. Indeed, while red light reduces photobleaching of human rhodopsin, dogs, unlike humans, are typically dichromatic with poor red discrimination [18].

Dogs not only have increased sensitivity to the brightness of light but also a higher critical flicker fusion frequency (CFFF) (the threshold at which an animal perceives flickering light as continuous rather than a series of flashes) than humans (80 Hz in dogs compared to 60Hz in humans) [19]. As a result, a video image that appears to be moving fluidly to humans may appear as a flickering set of images to dogs. The perception of such flickering may lead to symptoms associated with flicker vertigo, such as disorientation or impaired motor functions [8]. Although dogs' eyes are likely to be more sensitive to flickering or flashing lights than humans, it is unknown whether such sensitivity impairs their ability to perform physical or cognitive tasks. Working dogs can be exposed to flickering lights in their working environment (e.g., emergency siren lights or strobe lights used in the entertainment industry), thus it is important to understand the potentially detrimental effects of flashing visual stimuli on trained task performance in the field.

In addition to differences in the ways in which dogs and humans perceive light, marked anatomical and physiological differences related to hearing that are likely to produce differences in perception. For example, the outer ear of a dog can amplify sounds to a greater level than humans' (although there is more variation in dogs owing to their diverse ear shapes [9]), and many dogs are also able to hear sounds that are higher in pitch than humans (including ultrasound), but not those lower in pitch [20]. **In animals from** a range of taxa, exposure to sudden loud noises can result in an acoustic distractor response; a series of reflexive muscular contractions produced in response to acoustic stimuli, including blinking and postural tension, and a cortisol response [21]. Working dogs are exposed to many sudden changes in noise levels (explosives, construction machinery, traffic horns, helicopters etc.). Yet, the effects of

exposure to acoustic distractors on dogs' task performance are largely unknown, including whether repeated exposure to sudden loud noises generally has a response-decreasing (i.e., a habituating), or a response-increasing (i.e., a sensitising) effect on the physical and cognitive requirements of a performance task.

In the present study, we explored how changes in acoustic and light stimuli impact physical and cognitive performance in dogs. To do so, we trained pet dogs to navigate across a floor-ladder runway before completing a visual discrimination task. The accuracy and time taken to complete this task was measured before, during and after exposure to three stimuli: an acoustic distractor (85dB), strobe lighting (5, 10 & 15 Hz), and exposure to a bright white, red or blue light after visual adaptation to the dark (dark adaptation). Our experiment allowed us to gain insight into how changes to the acoustic and visual environment impact the execution of complex motor patterns and/or cognitive decision-making skills involved in task execution by dogs.

## Methods

### Overview

Pet dogs were trained to walk over a floor-ladder runway, then choose and enter one of two 'goal areas' based on the presence of a dot placed pseudo randomly on one of two computer screens (Fig 1, S1 & S2 Figs). Once dogs passed a training performance threshold, they were asked to complete the task in the presence or absence of three conditions: either '*strobe lighting*', an '*acoustic distractor*', or following exposure to a bright light; hereafter termed '*dark adaptation*'. Latency to start the trial, the speed with which the dog completed the trial, and goal choice accuracy were compared before, during and after exposure to each condition to

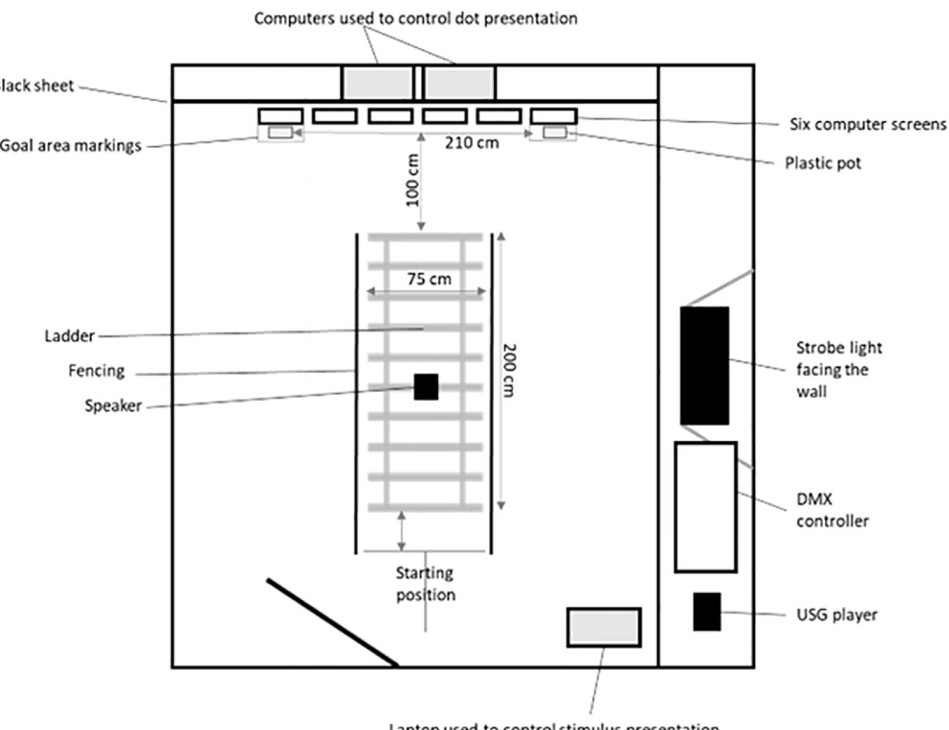

**Fig 1. Outline of the experimental setup of the testing room.**

assess the impact of stimuli exposure on performance. All work was approved by the relevant University Ethics Committee (References 2020–2127 and 2020–2328).

## Subjects

Twelve pet dogs, representative of various working dog types, were recruited through a database of volunteers at the University of Lincoln ('PetsCanDo', S1 Table) and were screened for a Canine Cognitive Dysfunction Rating (S4 Fig). The sample consisted of 8 females and 4 males with a mean age of 3.79 years (range 1 to 9 years old, standard deviation: 2.44, S1 Table). All owners provided informed consent documents to experiment on their pet dog.

## Experimental setting

The experiment was conducted in two connected rooms, separated by a door (S1 Fig). One room was a holding room, and the other was a testing room. Dogs were kept with a researcher in the holding room before entering the testing room to complete a trial. The holding room was 500 × 900 cm, with two shelves (120 & 100 cm high) in the centre of the wall at either end of the rectangular room, on which two Martin Harman Atomic 3000 LED lights were mounted facing into the room at an angle of 45˚. Installed onto the front of both lights were two Martin Harman Atomic Colours Colour Scrollers. On the ceiling, halfway down the long walls on either side, two Ledj Astra 12 quad parabolic aluminized reflector (PAR) lamps were mounted, facing down and into the room at an angle of 45˚. These lights were connected to the programmable ETC ColorSource 20 console (DMX controller), which was used to turn the lights on and off and control light colour during the dark adaptation condition (S1 Fig).

The testing room was 450 × 500 cm (Fig 1). A Vifa omni-directional ultrasonic speaker was mounted in the centre of the ceiling, connected to an Avisoft Bioacoustics USG Player 416H and controlled using an ASUS Zenbook UX433FA laptop running Avisoft RECORDER USGH software. A Martin Harman Atomic 3000 LED strobe light was mounted on a shelf at a height of 120cm facing the wall at an angle of 45˚ at the midpoint of the wall, connected to a programmable ETC ColorSource 20 console (Fig 1). At the far end of the testing room, on the floor, a row of six computer monitors (ViewSonic VA703B) were spaced 4 cm apart. The monitors were connected to two computers (Dell Optiplex 2080 desktop computer & HP EliteBook 840 G3 laptop) so that either a black screen (0,0,0), or a black screen with a white (255,255,255) dot (7 cm Ø) in the middle, could be presented using Microsoft PowerPoint. Directly in front of the two outer screens (Screens 1 and 6 when viewed from left-to-right) were two goal areas marked on the floor (38 x 29 cm). The distance between the two goal areas measured 177 cm, and the central point between them was 334 cm from the dogs' starting position. Directly in front of the dog was a foam ladder (200 × 75 cm) with ten rungs, which finished 100 cm before the screens. Along the sides of the ladder were two 30 cm high metal fences.

## Training procedure

Using shaping, dogs were initially trained to touch a white plastic circle (7 cm Ø) with their nose on the command 'touch' and given half a slice of hot dog (0.5 cm thick, **weighing approximately 5 grams**) as a reward. The circle was placed in various locations around the testing room, progressively further away from the dog, and eventually transferred to the front of the monitors. At this point, the plastic circle was replaced with an electronic dot of the same colour and dimensions on the screens. Once dogs were reliably approaching the dot from the other side of the room, they were trained to walk over the ladder using a luring procedure. This element of the task was then linked via a backwards chaining procedure into the routine of touching the electronic dot on the command 'touch.' A start position was then introduced, meaning

that dogs began each trial outside of the testing room (in the holding room), before being led into position in the testing room by an experimenter. Once in position (location marked on the floor (Fig 1)) the experimenter held them in a standing position by their harness or collar. The dot was present on the screen when dogs entered, but once in the starting position dogs were held in place for three seconds before being released and allowed to make a choice. From this point onwards, dogs were no longer required to touch the dot, instead they needed to get within 23 cm of the screen (inside the goal area marked on the floor). When dogs approached the correct screen, food was thrown into the goal area (Fig 1). If dogs approached the incorrect screen, no reward was provided, and they were collected by the experimenter and led outside into the holding room for another trial.

Dogs were trained in sessions of 16 trials, with 8 dots presented on each of the two screens in a pseudo randomised order. It was considered that dogs had successfully completed the training phase when they made a maximum of one mistake on three consecutive sessions of 16 trials (i.e., when the dog walked down the ladder and chose the correct goal area marked by the dot 47 out of 48 times). After dogs successfully completed the training phase, they were then habituated to wearing a Polar X10 heart rate monitor with Anagel ultrasound transmission gel. To get a reliable signal, the ultrasound transmission gel was worked into the dog's fur underneath the strap, and the fur was parted to expose the dog's skin (see S3 Table for heart rate data throughout the experiment).

**Trial 'drop-out,' sensitisation and dog welfare.** Dogs were considered to have 'dropped out' of the task when they i) did not complete the physical component of the trial (e.g., because they bypassed the ladder), ii) did not attempt/complete the task after 30 seconds from release, or iii) refused to participate in the trial on more than three occasions. Dogs were considered 'refusing to participate' when they did not initiate participation by leading the researcher into the testing room. If dogs did not initiate participation, they were not encouraged to enter the testing room or to start the task. These measures were taken to avoid stimulus sensitisation to the extent that an enduring behavioural response to a stimulus may have been produced. The light and acoustic distractor stimuli exposed to the pet dogs were at a level below which fear is likely to be exhibited, however this is very dependent on the response threshold of individual dogs [22]. After dogs dropped-out of the experiment, they were given unlimited time to explore the holding and testing rooms and were provided with food and play rewards. Participating dog owners only reported positive behavioural effects associated with their pet's participation in the experiment. No concerns of enduring behavioural changes in response to acoustic or light stimuli were reported.

**Testing procedure and stimuli used.** Each testing session comprised eight trials. The dot was pseudo-randomly displayed on screen one or six, such that on four trials the dot was presented on screen one and on four trials the dot was presented on screen six (this order of display was predetermined by a random number generator using the RAND function in Microsoft Excel). During every session of eight trials, dogs experienced one trial (randomly either Trial 3 or 4) that contained the dark adaptation, strobe or acoustic distractor stimulus, referred to as the 'stimulus trial'. The dark adaptation and strobe lighting conditions were counterbalanced for presentation order between dogs, with six dogs completing the dark adaptation condition first and six dogs completing the strobe lighting condition first. The presentation order of different coloured (red, white, blue) and frequency (5,10,15 Hz) stimuli was also counterbalanced within and between dogs to the best of our ability (S2 Table). The acoustic distractor condition was presented last as piloting showed that it was most likely to produce carryover effects.

*Dark adaptation condition.* Before each dark adaptation testing session, dogs were kept with a researcher in a dark room (the holding room, without windows or lights) for twenty

minutes. After twenty minutes, trials began, and during the stimulus trial the holding room was flooded with an intensely bright light for one minute. The light was either white (RGB: 255,255,255) (2400 lux), red (255,0,0) (610 lux) or blue (0,0,255) (1130 lux) (S2 Fig). After one-minute of bright light exposure, dogs were led by an experimenter into the testing room (low-level lighting; 2.5 lux) and manoeuvred into the start position. Dogs were kept in the start position for three seconds before being released to attempt the task on the command 'touch'. Dogs were kept in the dark with a researcher throughout the dark adaptation sessions. Each dog experienced six presentations of each colour stimulus, 18 sessions in total.

*Strobe lighting condition*. Before each trial, dogs were kept with a researcher in the holding room (lights turned on, ambient lighting approximately 350 lux) for three minutes, after which they were led into the testing room (2.5 lux), manoeuvred to the starting position for three seconds before being released to attempt the task on the command 'touch'. During the stimulus trial, a strobe light (with an intensity of 100,000 lux and a flash rate of either 5, 10 or 15 Hz) was presented as soon as the dog proceeded to move down the ladder runway. **For reference, 500 lux is equivalent to the brightness of a laptop screen, and 100,000 lux is equivalent to direct sunlight [23].** Dogs experienced two trials at each speed, creating 6 sessions in total. To reduce the risk of any cumulative effects from the stimulus on dogs an inter trial interval of one1 hour was used between each session.

*Acoustic distractor condition*. Before each trial, dogs were kept at ambient lighting (350 lux) in the holding room for 1 minute before being led into the testing room (2.5 lux) and manoeuvred into the starting position. During the stimulus trial an 85dB acoustic distractor was played as soon as the dog started to walk over the ladder obstacle. To reduce the risk of any cumulative effects from the stimulus on dogs an inter session interval of at least 1 hour was used. Each dog experienced a maximum of six sessions; however, if they refused to start the ladder on three subsequent trials, or showed any reluctance to re-enter the testing room their participation was terminated, and the dog was considered to have 'dropped-out' of this condition.

To ensure that the researcher (ELS) restraining the dog did not inadvertently influence the behaviour of dogs, ELS was unaware of the trial identity (i.e., whether the dot was on screen 1 or 6) prior to its onset, a second experimenter (CJE) controlled this. ELS also wore a cap to obscure her view of the location of the dot and headphones to reduce the sound.

## Behavioural data collection

Test sessions were recorded by four closed-circuit television (CCTV) cameras mounted on the ceiling in strategic locations (S1 Fig). Videos from the four cameras were synchronised and collated into a single video using the VSDC Free Video Editor software (S1 Fig). This video was then imported into the 'Behavioural Observation Research Interactive Software' BORIS software (version 7.9.7), and a continuous sampling technique was used to collect data on the following actions: a) when the dog was released by the experimenter; b) when their lead foot crossed the first rung of the ladder; c) when their last foot crossed the last rung of the ladder; d) when the stimulus was present (if applicable); e) when the dog entered the goal area or when the dog did not complete the trial fully. A trial was considered incomplete if the dog either by-passed the ladder to access the goal area or did not move from the start position within 30 seconds of their release.

Behavioural data were collected during the interval spanning the time from when the dog was placed in the start position to the time that the dog made a choice. Latency to start the trial was measured as the time between when the experimenter released the dog, and when the dog's lead foot crossed the first rung of the ladder. Time taken to run down the floor-ladder was measured as the time between when the dog's lead foot crossed the first rung of the ladder

and when the last foot crossed the last rung of the ladder. A dog's response was coded as 'correct' if it went to the goal area with the electronic dot displayed on the screen above it, and 'incorrect' if it went to the goal area with a blank black screen above it. If dogs 'turned too soon' (i.e., if a dog approached the screen displaying the dot, but turned before their head entered the goal area in front of the screen), their accuracy was recorded (coded as correct), but the trial duration was not. "No-choice" was coded if dogs did not make a clear selection within 30 seconds.

## Statistical analyses

All statistical analyses were conducted in R 4.1.2 [24]. Two researchers (ELS and CJE) coded behavioural events from video recordings of the dogs' trials using BORIS. To ensure the coding was repeatable between the researchers 16.6% of total recordings were coded by two researchers using the same BORIS ethograms. The repeatability of temporal (state/point) events (e.g., latency to start the trial and the time taken to walk down the ladder) was analysed using a Pearson's correlation coefficient test, and the repeatability of binary, categorical events (i.e., coding of correct or incorrect goal choice) was assessed by comparing the percentage of times both researchers coded the same categorical event. Coding repeatability of temporal events in BORIS was high ($r = 0.987$, $p<0.001$) and the repeatability of binary, categorical events (goal choice accuracy) was 100%. **We log transformed data on "*latency to start the trial*" and "*time taken to complete the task*" to improve model diagnostics.**

We ran linear mixed models (LMMs) to compare the relative impacts of dark adaptation, strobe lighting and the acoustic distractor on the task performance of each dog using the lmerTest package [25] in R. In our first model, we used latency to start the trial (seconds) as the dependent variable. We included condition (dark adaptation, strobe lighting or the acoustic distractor), session type (before and after exposure to the stimulus) and their interaction as fixed effects. We also included trial as a fixed effects to control for the potential effects of repeated exposure as the experiment progressed. 'Dog ID' (the name of the pet dog) and 'Session' were included as random effects. Our model allowed us to simultaneously compare the relative impact of each condition on performance, and how each condition affected individual performance (i.e., how performance varied before, during and after exposure to each stimulus). Post hoc pairwise differences were tested using the emmeans package in R. We also tested the effect of strobe frequency (5Hz, 10Hz, 15Hz) and the colour of the light (white, red, blue) on the latency to start the trial during exposure to the stimulus in a separate LMM. Here, only Dog ID was included as a random effect.

In our second model, we investigated whether the time taken to complete the practical element of the task (i.e., the time taken (seconds) to walk down the ground-ladder) (the dependent variable), was affected by exposure to the three conditions. As before, we included stimulus, session type (this time before, *during* and after exposure to the stimulus) and their interaction as fixed effects, trial number, and session number as fixed effects, with 'Dog ID' and 'Session' as random effects. Again, we followed this test by exploring the effect of strobe frequency and the colour of the light on the time taken to complete the practical element of the task.

Our third aim was to assess whether choice accuracy (i.e., the proportion of trials for which the dog correctly entered the 'goal area' indicated by the presence of a digital dot), was affected by the three conditions. We ran a generalised linear mixed-effects models (GLMMs) using the glmer function of the lmer4 package in R, with the 'family' option used to specify a binomial distribution for the dependent variable 'choice accuracy' (i.e., a correct or incorrect goal area choice). Fixed and random effects included in these models were the same as described previously.

After data were log transformed, we conducted Shapiro-Wilk and non-studentized Breusch-Pagan analyses using the R packages 'olsrr' and 'lmtest' to confirm that the residuals of our models were normally distributed and homoscedastic [26]). Models were built on the biological variables of interest. A *post hoc* power analysis was conducted using the R package SIMR in to assess whether our sample size (*N* = 12 dogs, with 48, 72 and 48 observations per dog during the strobe, dark adaptation and acoustic stimuli condition, respectively) was adequate for the magnitude of effects observed to be statistically reliable [27]. This power analysis suggested that a sample size of n ≥ 9 was required to detect an effect of our fixed factors on dogs' task performance with 83% power at a 0.05 significance level, thus our sample size (*N* = 12 dogs) was more than adequate. Sessions for which dogs did not complete every trial (i.e., drop-out sessions) were not included in our analyses. 'Session number' did not impact task performance in any of our analyses. Therefore, to avoid biasing our dataset (by omitting data from the dogs that sensitised to the acoustic distractor), we used a 'last observation carried forward' technique to replace missing follow-up observations with previous observations. This technique was suitable to replace the missing data as there were no trends for dogs to change their performance across different sessions (see Results section). Our results did not differ when uncompleted sessions were omitted from the data set.

## Results

### Trial 'drop-out' rates

During the strobe and dark adaptation conditions, all dogs completed the task (i.e., chose a goal area, whether incorrect or correct) in every trial. However, during the acoustic distractor condition, two dogs (Max and Lilly) 'dropped out' of the task (i.e., did not complete the physical component of the trial because they bypassed the ladder, did not attempt/complete the task after 30 seconds from release, or refused to participate in the trial on more than three consecutive occasions) during the first session. Three dogs (Logan, Lyra and Poppy) dropped out during the third trial, three dogs (Arlo, Hunter and Ava) dropped out during the fourth trial and one dog (Sky) dropped out during the fifth trial. Only two dogs (Hector and Alexis) finished the acoustic distractor condition by completing the six trials (See S1 Table for details of each dog participant). This resulted in a 'drop-out' rate of 77.7% for the acoustic distractor condition (i.e., 9 out of 11 dogs did not complete the acoustic distractor condition), and 47% of acoustic stimuli trials were not attempted. One dog (Pan) did not take part in the acoustic distractor or strobe condition due to underlying health issues and two dogs (Ava and Hunter) did not take part in the light conditions due to time-constraints imposed by the owners (S1 Table).

### Latency to start the trial

Latency to start the trial was significantly longer after exposure to the acoustic distractor stimulus compared to before exposure ($t$ = 6.090, $p<0.001$). However, the visual stimuli (strobe and dark adaptation) had no effect on latency to start the trial ($p>0.20$ for all pairwise comparisons) (Fig 2). Dogs took longer to start the trial after being exposed to the acoustic distractor stimulus compared to the exposure to the strobe or dark adaptation stimuli (post-hoc comparisons: $t$ = 2.834, $p$ = 0.052, $t$ = 2.842, $p$ = 0.051, respectively). 'Latency to start the trial' could not be assessed 'during' stimulus exposure as the stimulus only commenced after the dog commenced the trial. Trial number or session number were not associated with latency to start the trial, and we found no effect of specific light colour on latency to start the trial during the trials. The model explained a total of 56.8% of variance in the data on dogs' latency to start the trial, however, most variance (49.4%) was explained by individual differences between dogs.

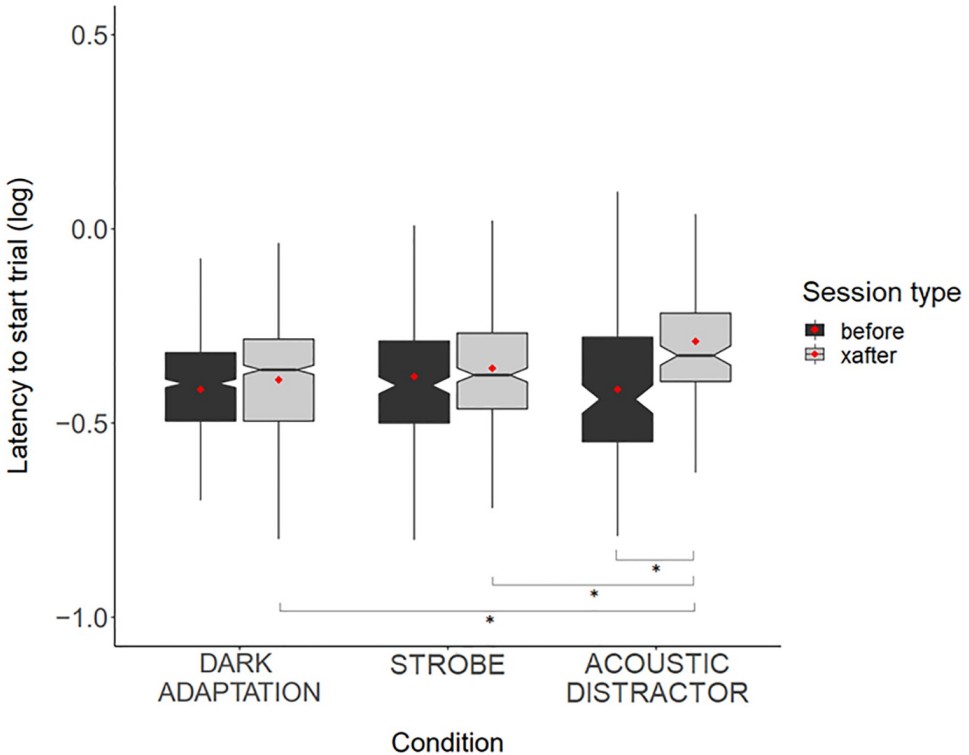

**Fig 2. Boxplot showing the latency to start the trial before and after exposure to the dark adaptation, strobe, and acoustic distractor.** Red dots indicate mean values. Notches indicate median values, boxes represent the 25th and 75th percentile of data, and 'whiskers' represent the minimum and maximum data range values. (*) refers to statistically significant differences.

## Time taken to perform the task

We detected a significant effect of stimulus condition on the time taken to walk down the floor ladder which was dependent on session type; dogs took less time to walk down the ladder before, versus during and after exposure to the acoustic distractor (post-hoc comparisons: $t = -6.622$, $p < .0.001$, $t = -5.232$, $p<0.001$, respectively) (Fig 3). Dogs took less time to walk down the ladder before and after versus during exposure to the strobe light (post-hoc comparisons: $t = -4.853$, $p <0.001$; $t = 3.280$, $p = 0.0292$, respectively), however, dark adaptation did not impact the length of time taken to walk down the floor ladder ($p>0.8$ for all pairwise comparisons). Dogs took longer to walk down the ladder during exposure to the acoustic distractor compared to during exposure to the dark adaptation ($t = -4.694$, $p = 0.001$) and strobe stimulus ($t = 5.410$, $p = <0.001$). Trial number and session number did not impact time taken to walk down the floor ladder and we found no effect of specific light colour on the duration to walk down the floor ladder (Fig 3). The model explained a total of 72.4% of variance in the data on the time taken to walk down the floor ladder, however, most variance (68.3%) was explained by individual differences between dogs.

## Choice accuracy

There was a significant effect of stimulus condition on choice accuracy (Fig 4), and this was also contingent on session type; dogs were less accurate during versus before exposure to the acoustic distractor and were also less accurate after versus before exposure to the acoustic

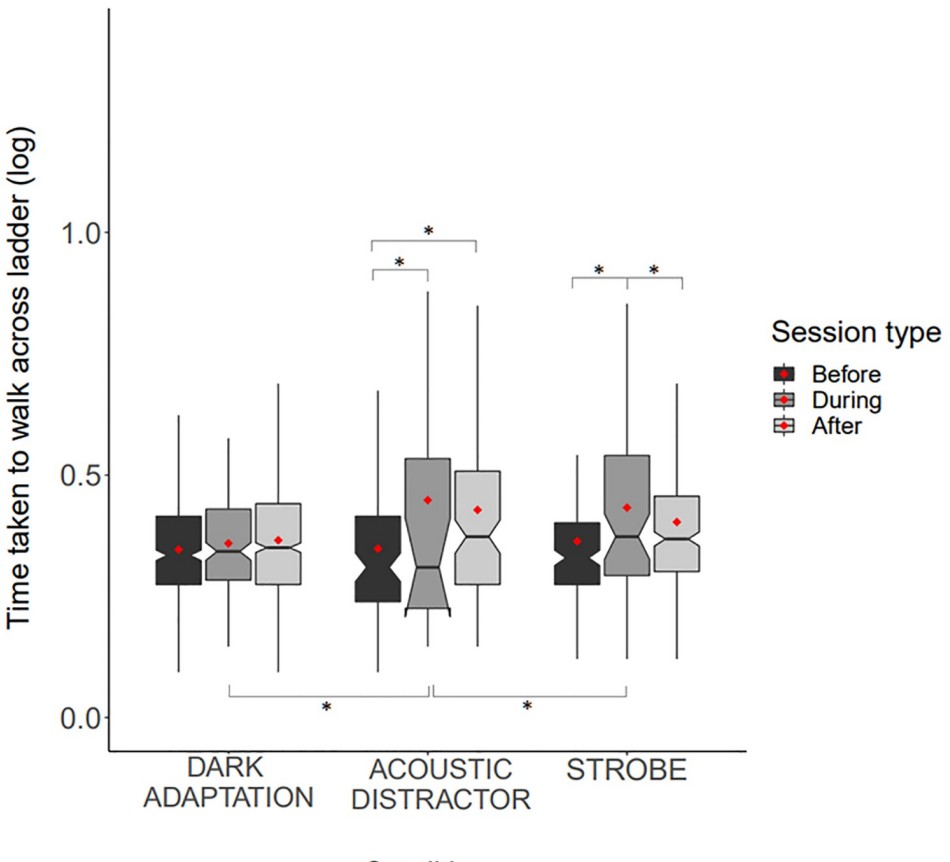

**Fig 3. Boxplot showing the time taken to walk across the floor ladder, before, during and after exposure to the dark adaptation, acoustic stimulus, and strobe stimuli.** Red dots indicate mean values, notches indicate median values, boxes represent the 25th and 75th percentile of data, and 'whiskers' represent the minimum and maximum data values. (*) refers to statistically significant differences.

distractor (post-hoc comparisons: $z = 4.195$, $p < 0.001$; $t = 3.371$, $p = 0.021$, respectively). The dark adaptation and strobe condition did not impact accuracy. Dogs were less accurate during exposure to the acoustic distractor compared to during exposure to the dark adaptation and strobe condition (post-hoc comparisons: $z = 4.408$, $p < 0.001$; $t = 3.575$, $p = 0.016$, respectively). Dogs were also less accurate after exposure to the acoustic stimulus compared to after exposure to the dark adaptation and strobe stimulus (post-hoc comparisons: $t = 5.453$, $p < 0.001$; $z = 4.560$, $p < 0.001$, respectively). Accuracy before exposure to the acoustic distractor stimulus did not differ from accuracy before exposure to the strobe or dark adaptation stimuli ($p > 0.28$ in all comparisons).

Choice accuracy decreased across the acoustic distractor sessions (i.e., dogs tended to sensitise to this stimulus across the experiment) (session number: estimate = -1.821 (Std Error = 0.523), $z = -3.126$, $p < 0.002$). This decrease in accuracy was greater in dogs that dropped out earlier (session number*drop-out stage: estimate = 0.409 (Std Error = 0.126), $z = 3.327$, $p = 0.001$). It is notable that no dogs exhibited enduring behavioural changes to light or acoustic stimuli, see the paragraph titled 'Trial drop-out. sensitisation and dog welfare'. There was no difference in choice accuracy between the different coloured lights (white, red, blue) in the dark adaptation condition, and no differences based on the strobe frequency. The

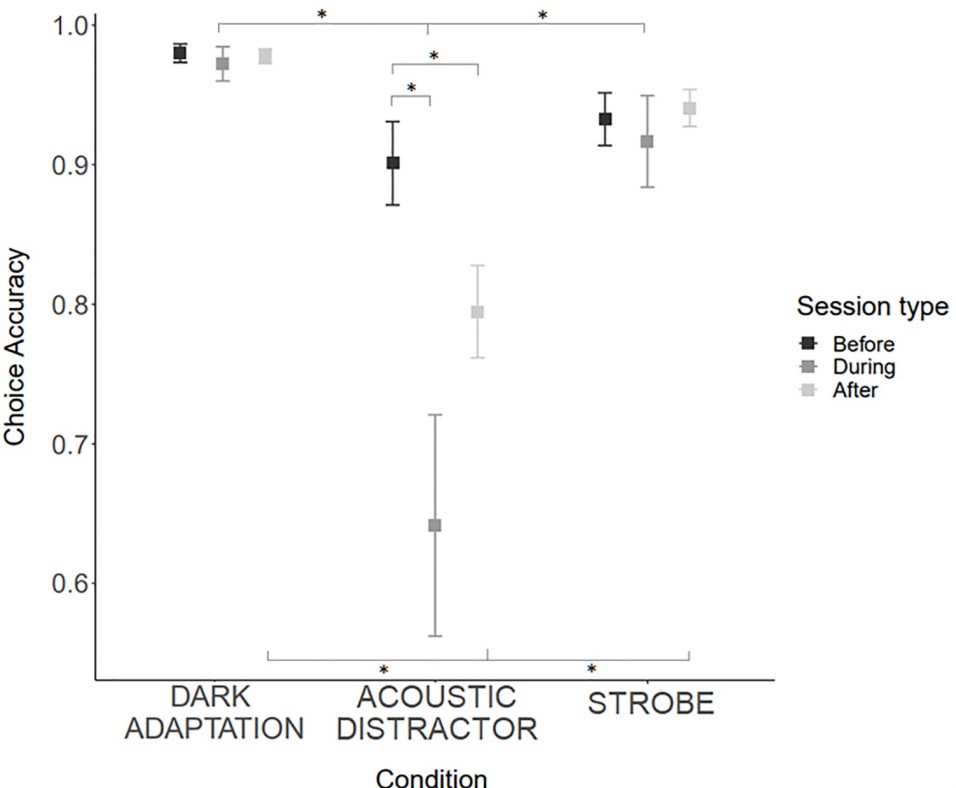

**Fig 4. Average choice accuracy (with standard deviation bars) before, during and after exposure to the dark adaptation, acoustic stimulus, and strobe stimuli.** (Note: a boxplot was not appropriate for data visualisation here due to the binary nature of 'choice accuracy'). Error bars represent the standard deviation of data, and filled squares represent the median data values. (*) refer to statistically significant differences.

model explained a total of 30.6% of variance in the accuracy data on dogs,' 7.1% was explained by individual differences between dogs, and the remaining variance by the fixed factors.

## Discussion

The aim of our study was to examine the impact of sudden changes in sound and light on dogs' physical and cognitive performance. We found that exposure to an acoustic distractor and to a lesser extent exposure to strobe lighting lengthened the time taken to complete a physical task component. Exposure to an acoustic distractor also lengthened the time taken to start the task. Dark adaptation (i.e., dark vision recovery after photobleaching) did not affect task performance neither did light colour (red, blue, or white), or strobe frequency. Exposure to the acoustic distractor was the only stimulus that impacted dogs' cognitive performance; choice accuracy being significantly lower during and after versus before exposure to this stimulus. While all dogs completed the strobe and the dark adaptation trials, dogs failed to complete the acoustic distractor trials 47% of the time, and 72% of participants (9 out of 11 dogs) 'dropped-out' of this experimental condition early. The dogs that dropped-out of the experiment seemed to sensitise to the acoustic distractor, while the two dogs that completed the acoustic distractor sessions tended to habituate to this stimulus. These results highlight the significance of inter-individual variation in response behaviours to distractive stimuli and suggest that in addition to finding acoustic stimuli distracting, many dogs also find them aversive.

Although dark adaptation is reported to be twice as long in dogs as it is in humans [11], recovery from the effects of photo bleaching following exposure to red, blue, or white light did not affect dogs' physical or cognitive task performance in our experiment. The time taken (~15 seconds) to lead the dog into the testing room, put them in the start position and wait for two seconds before releasing, is unlikely to have allowed the dogs' rods to regenerate to a sufficient level to perform the task, given that it has been suggested that dogs' eyes need around one hour to fully recover [13]. Instead, it is likely that the changes to dogs' vision following photo-bleaching were not adequate to induce performance deficits in our task. The researcher who handled the dog, and therefore experienced the same conditions, reported negative afterimage effects after exposure to red and blue light, in which, the testing room appeared to be the opposite colour on a colour wheel (for example, exposure to blue light produced a red-orange colour distortion). Since afterimages are produced by the brain adapting to a reduced signal level resulting from over-stimulated photoreceptors, it is possible that dogs also experienced negative afterimages in this condition, but again, these changes were not sufficient to impair their performance.

While dark adaptation did not impair physical or cognitive aspects of dogs' task performance, strobe lighting negatively affected the physical aspect of the task (walking down a floor ladder). This finding is consistent with the idea that, like humans, strobe lighting (ranging from 5 to 15Hz) artificially reduces dogs' visual frame rate, impairing their perception of continual motion, making it harder to traverse an obstacle. This interpretation is supported by a previous experiment conducted by [28] who found that cats placed their feet with less accuracy across a cluttered terrain in the presence of low-frequency strobe lighting. In a complementary study, [29] showed that this difference was not due to an impairment in visual acuity, as the cats performed better in lower light levels than that created by the strobe lighting. A significant methodological difference between these experiments and the current one is that obstacle spacings for the cats were unpredictable, whilst in our experiments, the rungs on the ladder were equidistant from each other, making it potentially easier to cross. It is therefore likely that dogs' physical performance would be more heavily impacted by strobe lighting in real-world, variable terrain situations.

The acoustic distractor was the only stimulus to significantly impair choice accuracy, suggesting that dogs found this stimulus to be the most cognitively distracting. The negative impacts of unexpected sounds on memory have been well-studied in the human literature [30], with many explanations claiming that sound disrupts serial recall because it captures attention and therefore removes processing resources from the task at hand [31]. Since the visual stimuli did not impact choice accuracy, and because the onset of the acoustic distractor was after dogs had been held in the start position in view of the electronic dot for two seconds, it is unlikely that reductions in choice accuracy were caused by dogs failing to encode the location of the electronic dot, and therefore the correct goal location. Instead, the maintenance of the dot's position in dogs' short-term memory (<30 seconds) was disrupted by the attentional capture of the acoustic stimulus. Certainly, in humans, task-irrelevant distractors interfere with memory retrieval [32]. To date, the current study represents the only known investigation into the effects of task-irrelevant sound and light distractors in dogs. However, [33] has previously shown that the presence of task-relevant items impairs dogs' performance on a visual search task. Alternatively, the incentive-based motivation for task performance may have been dampened by dogs' perception of the acoustic stimulus as aversive. Indeed, most dogs 'dropped out' of the acoustic stimuli tasks (i.e., did not initiate participation in the task on more than three occasions). Further, the dogs that did 'drop-out' of tasks took progressively longer to start the task on repeated exposure to the acoustic stimulus, indicating that this stimulus may have been aversive in addition to task-irrelevant in most dogs [34].

It is well known that many dogs show fearful responses to sudden loud sounds [34], and many predator deterrents, for example those designed to protect livestock from coyotes (*Canis latrans*), have utilised acoustic distractors in their designs. Interestingly, many of these deterrents couple acoustic stimuli with strobe stimuli; yet whether a combination of these stimuli exert additive physical/cognitive impairments is unclear, and is an avenue for future work [35]. Notably, not all dogs sensitised with repeated exposure to the acoustic distractor (two dogs desensitised) in our study. Thus, another productive line of investigation would be to unravel the intrinsic factors (e.g., an individual's level of noise aversion, cognitive load, age, or arousal level [36–38] which make a dog more likely to sensitise rather than desensitise to sudden loud sounds, as these factors could be used as selection criteria for dogs working in relevant environments [39]. It would also be valuable to identify the extrinsic factors (e.g., variation in reward type, visual and/or acoustic protection, and habituation training regimes) that are most effective in mitigating performance impairments in the presence of extraneous distractors.

## Study limitations

While our sample size was relatively small, *N* = 12 dogs, power analyses confirmed that it was more than adequate to detect statistically meaningful effects of the size seen here. Inter-individual variation in task performance was high and explained the greatest proportion of variation in our task performance data. Nonetheless, the strobe and the acoustic stimulus tended to affect individual performance in a consistent manner–such that significant, negative performance effects were detectable. Our study reveals how different distractors impair task performance in dogs. An important next step is to further assess this in relation to the challenges working dogs may face. Working dogs are exposed to multiple distractors from different modalities, which may occur simultaneously. It would therefore be valuable to assess how a combination of different stimuli may impact dogs' task performance, and future work should identify the specific (or specific combination of) stimuli that dogs find most distracting, to develop effective strategies to mitigate this. Our sample size was not large enough to explore how differences in the anatomical features of dogs may have affected their responses; for example, it would be useful to useful whether prick eared dogs (characteristic of police dogs) are more sensitive to acoustic distraction than lop eared dogs. Interestingly many gun-dog breeds have lopped ears and more often hairy ear canals that might serve as a form of auditory baffle [9]. This is an important avenue for future work.

## Conclusion

Our study has highlighted the potential for strobe and acoustic distractors to interfere with physical and cognitive aspects of dogs' task performance. Our findings have important implications for handlers working with dogs in environments containing sudden changes in sound and/or rapid, recurrent changes in light-levels. Our results indicate that without effective distractor response training or habituation training, sudden changes in noise and light levels are likely to impede cognitive and physical task performance in working dogs, and repeated exposure may cause these effects to be amplified.

Our study suggests that habituating dogs to dazzling lights (i.e., dark adaptation) is unlikely to improve task performance, and that time and resources may be better spent desensitising dogs to strobe (or flickering) lights and acoustic stimuli. Indeed, the gradual introduction of loud noises to the training of improvised explosive device (IED)-detection dogs has been shown to improve the performance of IEDs-detection dogs in the field [40, 41] k. Additionally, repeated exposure to visuospatial distractors has been shown to improve performance and

information filtering mechanisms in humans [42]. Thus, incorporating strobe lighting into the training regimes of relevant working dogs could also help to mitigate the performance impairments associated with this stimulus (however, more work is needed to effectively inform such regimes). Notably, the implementation of such desensitisation programmes should be conducted after the initial stages of training (once criterion has been met), as exposure to distractors during the initial stages of training may hinder the rate of task learning in dogs [43].

Future work should aim to identify the morphological or behavioural characteristics that are associated with dogs' sensitivity to light and sound stimuli. Additionally, testing whether performance impairments could be reduced by equipping dogs with hearing or visual protection would also constitute a useful next step (e.g., [44, 45]). Finally, because handler behaviour can affect dogs' task performance [46, 47], it would be useful to assess the effects of sound and light on the combined performance of the dog and handler *team*, to improve task performance.

## Supporting information

**S1 Fig. The experimental arena.** Outline of the experimental arrangement of the testing room and waiting room.
(DOCX)

**S2 Fig. Dark adaptation condition.** Ater 20 minutes spent in low lighting, dogs were subject to a room illuminated with either a) a white light, b) a red light, or c) a blue light, for one minute before being asked to perform the test task in low lighting. The rooms were lit by two Martin Harman Atomic 3000 LED light and two Ledj Astra 12 quad parabolic aluminized reflector (PAR) lamps.
(DOCX)

**S3 Fig. Test recording sessions.** Test sessions were recorded by closed-circuit television (CCTV) cameras mounted on the ceiling in strategic locations. All cameras streamed to a HIK-VISION digital video recorder which recorded them as separate videos. The cameras were set to capture the following data: Camera 1 (S3 Fig, top left) captured an overview of the whole trial, including which screen the dot was presented on; Camera 2 (S3 Fig, bottom left) was positioned directly above the dog's head to allow the coder to determine when the dog was released and when they started the ladder; Camera 3 (S3 Fig, top right) was used to determine the moment when the dog entered the goal area, and their choice was recorded; Camera 4 (S3 Fig, bottom right) was added to view the majority of the waiting room, including the door to the testing room. An example of a screenshot taken from cameras 1, 2, 3 and 4 (anti-clockwise from the top left).
(DOCX)

**S4 Fig. Canine Cognitive Dysfunction Rating (CCDR) scale.** Canine Cognitive Dysfunction Rating (CCDR) Scale questionnaire used, graphically modified from Salvin et al., (2011).
(DOCX)

**S1 Table. Participant recruitment.** Twelve pet dogs were recruited for our study. Requirements for recruitment were that dogs i) were older than 1 year, but not older than 10 years, ii) were in good health (including no visual deficits), iii) did not reach the threshold for presence of cognitive dysfunction as assessed using the canine cognitive dysfunction rating scale (Salvin et al., 2011), iv) were comfortable being handled by a stranger, v) were not food aggressive, vi) were at ease in the laboratory environment, and vii) motivated to train. The name, breed, sex, sexual status, age, ear type and eye colour of the dogs who took part in the experiment. 'GSD'

= German Shepherd, 'NSDTR' = Nova Scotia Duck Tolling Retriever, 'x' refers to a cross breed.
(DOCX)

**S2 Table. Stimulus presentation order.** The presentation order of red, white and blue light, and 5Hz, 10Hz and 15Hz strobe light frequency.
(DOCX)

**S3 Table. Physiological data.** Heart Rate data was recorded continuously during every trial, and was later averaged before, during and after exposure to the stimulus at the point when the dog was released, when it had proceeded halfway down the ladder, and as it approached the goal area. Inter-beat interval data was recorded using a Polar X10 heart rate monitor in conjunction with the HR and HRV Logger app. From this, the dogs' heart beats per minute (bpm) were calculated across a three second rolling average time frame along with its associated time-stamp. This information was then imported into the BORIS software and synchronised with the videos. Heart rate data was averaged across each session. Mean heartrate (beats per minute (bpm)) and standard deviation (±) when dogs were released, started the ladder and approached a goal area in the dark adaptation, strobe lighting and sound condition, on the trial before, during and after a stimulus was presented.
(DOCX)

# Acknowledgments

The authors would like to sincerely thank the dog owners who brought their pets; Max, Lilly, Logan, Lyra, Poppy, Arlo, Hunter, Ava, Sky, Alexis and Hector to partake in this study.

# Author Contributions

**Conceptualization:** Carla J. Hart, Anna Wilkinson, Carl Soulsbury, Victoria F. Ratcliffe, Daniel S. Mills.

**Data curation:** Elizabeth L. Sheldon, Carla J. Hart, Anna Wilkinson, Carl Soulsbury, Daniel S. Mills.

**Formal analysis:** Elizabeth L. Sheldon, Carla J. Hart.

**Funding acquisition:** Carla J. Hart, Anna Wilkinson, Carl Soulsbury, Daniel S. Mills.

**Investigation:** Elizabeth L. Sheldon, Carla J. Hart, Anna Wilkinson, Carl Soulsbury, Victoria F. Ratcliffe, Daniel S. Mills.

**Methodology:** Elizabeth L. Sheldon, Carla J. Hart, Anna Wilkinson, Carl Soulsbury, Victoria F. Ratcliffe, Daniel S. Mills.

**Project administration:** Elizabeth L. Sheldon, Carla J. Hart, Anna Wilkinson, Carl Soulsbury, Victoria F. Ratcliffe, Daniel S. Mills.

**Resources:** Elizabeth L. Sheldon, Carla J. Hart, Daniel S. Mills.

**Supervision:** Elizabeth L. Sheldon, Carla J. Hart, Anna Wilkinson, Carl Soulsbury, Daniel S. Mills.

**Validation:** Elizabeth L. Sheldon, Carla J. Hart.

**Visualization:** Elizabeth L. Sheldon, Carla J. Hart.

**Writing – original draft:** Elizabeth L. Sheldon, Carla J. Hart.

**Writing – review & editing:** Elizabeth L. Sheldon, Carla J. Hart, Anna Wilkinson, Carl Soulsbury, Victoria F. Ratcliffe, Daniel S. Mills.

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
