## [Decision Letter · Decision Letter 0]

8 Oct 2023

PONE-D-23-15303Working dogs in dynamic on-duty environments: the impact of dark adaptation, strobe lighting and acoustic distraction on task performancePLOS ONE

Dear Dr. Sheldon,

Thank you for submitting your manuscript to PLOS ONE. After careful consideration, we feel that it has merit but does not fully meet PLOS ONE’s publication criteria as it currently stands. Therefore, we invite you to submit a revised version of the manuscript that addresses the points raised during the review process.

First of all I want to apologize on behalf of the journal for the long review process. Your submission was first attributed to a different editor, who unfortunately failed to secure expert reviewers. When the paper was reassigned to me, some time had already passed, and despite my best efforts it has been surprisingly difficult to find reviewers. While I usually try to avoid acting both as editor and reviewer, in the interest of not keeping you waiting any longer, I decided to do so here, as the paper is within my own area of expertise. You will find my reviewer comments below, together with those of the two independent reviewers. When revising the paper, please focus on the comments from Reviewer 2 and the Editor.

We look forward to receiving your revised manuscript.

Kind regards,

I Anna S Olsson, Ph.D.

Academic Editor

PLOS ONE

Journal Requirements:

Additional Editor Comments:

Editor review comments

This paper reports a study of how dogs’ performance in a cognitive task is affected by three types of sensory distraction: noise, strobe lighting and dazzling coloured light. The participating dogs are owner volunteered companion dogs, whereas the context for which the study is meant to be informative is working dogs. The study seems to be carefully designed and carried out, although the small sample size is a limitation.

Please see the following list of required revisions:

INTRODUCTION

Line 90 Please change to “In animals of/from a range of taxa”.

Line 95 Please consider removing “potential”, as here you are measuring the real effect.

METHODS

Line 163 Please indicate the size of the reward in grams.

Line 199 It would be helpful if you could give a similar indication of equivalence for the light strength as you do for the sound volume.

STATISTICAL ANALYSES

The second sentence refers to testing model residuals. Presumably this is done after there is data to be analysed. Shouldn’t this sentence then come later in the text? Lines 283-292 are all about coding behavioural events from the videos and testing for inter-observer agreement, which I understand is the process through which data are generated for statistical testing using LMM. This suggests to me that the information on model residuals should only come after the coding and agreement testing have been presented.

RESULTS

Lines 400-401 It is not clear to the reader what a carry-over effect would be here. Carry over from which situation to which situation? Also, please justify why it is relevant to refer to an effect which is not statistically significant.

DISCUSSION

Please add a section on the limitations of the study. This should address the small sample size, the large inter-individual variation (according to the statistics reported e g on lines 355-356 and 377-378 this accounted for almost all the variance explained by the model) and the potential for inferring from the study to actual working dog challenges. Also address, either here or in the relevant part of the Results section, how you dealt in the statistical analysis with the high drop-out rates.

In the final paragraph of the Discussion, you reflect on the implications for how working dogs should be trained. This is a relevant observation, but it should be complemented with a reflection on how dogs are trained in professional settings presently. I would be surprised if this aspect is not taken into account at least to some extent in existing professional dog training programs. There may not be scholarly papers, but there must be other sources of information. A quick search turned up a one-hour webinar from the US https://www.youtube.com/watch?v=nfB2L_NldA8 on how to manage distractions in police dog work. It may be that the easily available information does not address distractions of the type you have tested for, but you need to provide some evidence of how you have searched for this information.

REFERENCES

Please carefully check the references in the text against the reference list. The order in the list should correspond to the order in which the references appear in the text. For example, he second reference in the text is listed as the third in the list. Please also check the formatting of the references. Journal titles are sometimes capitalized, sometimes not, even for the same journal. Conference references are incomplete (should include location and date). Please see sample references here https://www.nlm.nih.gov/bsd/uniform_requirements.html

When referring to a source which is not a peer-reviewed paper, it should be clear from the text what the source is. When you use a company reference to the specifics of a piece of equipment, this is evident. But you also use industry references as sources for information on dogs’ hearing threshold, e g line 211. Is this the best source for this information?

Reviewers' comments:

Reviewer's Responses to Questions

**Comments to the Author**

1. Is the manuscript technically sound, and do the data support the conclusions?

Reviewer #1: No

Reviewer #2: Yes

2. Has the statistical analysis been performed appropriately and rigorously? 

Reviewer #1: No

Reviewer #2: Yes

3. Have the authors made all data underlying the findings in their manuscript fully available?

Reviewer #1: Yes

Reviewer #2: Yes

4. Is the manuscript presented in an intelligible fashion and written in standard English?

Reviewer #1: Yes

Reviewer #2: Yes

5. Review Comments to the Author

Reviewer #1: The study is interesting, but it is too preliminary and too small. The methods lack rigor and do not meet the high standards of this journal.

It also lacks rigor.

The extremely small sample and too basic info disallow me to accept the paper in this journal with high standards.

I suggest the authors augment the data to for example 40, 50 dogs or more and then reanalyze the data and resubmit their paper to PLOS One.

Reviewer #2: This study is a valuable contribution to the literature and clearly has practical relevance for working dog researchers and handlers. The findings are logically presented, analysed, and discussed.

One limitation of the work is that only 12 dogs participated and a considerable amount of inter-individual variability would be expected for this effect. However, this is a common limitation for experiments involving time-intensive training and testing and fortunately this sample of dogs did demonstrate a range of possible responses. I am satisfied that the conclusions are informative and not over-stated.

I also commend the researchers for the welfare measure of allowing dogs to initiate their own participation.

Minor comments:

I would be interested to know the actual breeds of participating dogs (rather than just “working dog types”). Individuals’ sensitivity to visual and acoustic stimuli may at least partly be determined by breed.

Under the heading “stimuli used” lines 190-199, it is not immediately clear which room each type of stimuli were presented in (e.g., I was unsure in this section what the lighting conditions were for the dark adaptation condition in the holding vs testing room). This is expanded upon later, line 236 onwards, but clarity may be improved by including this piece of information in the first instance.

6. PLOS authors have the option to publish the peer review history of their article (what does this mean?). If published, this will include your full peer review and any attached files.

Reviewer #1: No

Reviewer #2: No

---

## [Author Response · Author response to Decision Letter 0]

21 Nov 2023

Response to Reviewers

Working dogs in dynamic on-duty environments: the impact of dark adaptation, strobe lighting and acoustic distraction on task performance

Article: PONE-D-23-15303

Dear the Editor and two Reviewers of our manuscript, 

We would like to thank you for your detailed assessment of our work and the thoughtful suggestions that have greatly improved the quality of our manuscript. We have responded to each of your comments below, and numbered them accordingly (e.g., comment 1, response 1). Edits made in the resubmitted manuscript are highlighted with three asterisks (***). Edits in the manuscript are highlighted in bold.

On behalf of the authors, thank you again.

EDITOR REVIEW COMMENTS

Editors overview: This paper reports a study of how dogs’ performance in a cognitive task is affected by three types of sensory distraction: noise, strobe lighting and dazzling coloured light. The participating dogs are owner volunteered companion dogs, whereas the context for which the study is meant to be informative is working dogs. The study seems to be carefully designed and carried out, although the small sample size is a limitation. Please see the following list of required revisions:

INTRODUCTION

Comment #1: Line 90 Please change to “In animals of/from a range of taxa”.

Response #1: Edited

Comment #2: Line 95 Please consider removing “potential”, as here you are measuring the real effect.

Response #2: Edited

METHODS

Comment #3: Line 163 Please indicate the size of the reward in grams.

Response #3: Thank you, we have now edited this to:

*** ‘half a slice of hot dog (0.5 mm thick, weighing approximately 5 grams) as a reward.’

Comment #4: Line 199 It would be helpful if you could give a similar indication of equivalence for the light strength as you do for the sound volume.

Response #4: We have now provided a similar indication of light levels in ‘lux’ as we did for sound levels in decibels:

*** ‘For reference, 500 lux is equivalent to the brightness of a laptop screen, and 100,000 lux is equivalent to direct sunlight (The Engineering Toolbox, 2004).’

Citation: The Engineering ToolBox [Internet]. Illuminance - Recommended Light Levels. (2004). Available from: https://www.engineeringtoolbox.com/light-level-rooms-d_708.html [cited Oct 2023]

STATISTICAL ANALYSES

Comment # 5: The second sentence refers to testing model residuals. Presumably this is done after there is data to be analysed. Shouldn’t this sentence then come later in the text? Lines 283-292 are all about coding behavioural events from the videos and testing for inter-observer agreement, which I understand is the process through which data are generated for statistical testing using LMM. This suggests to me that the information on model residuals should only come after the coding and agreement testing have been presented.

Response #5: We agree that the positioning of this sentence was not logical. Per the editor’s suggestion we have now moved this sentence to the ‘statistical analysis’ section of our manuscript, 

RESULTS

Comment #6: Lines 400-401 It is not clear to the reader what a carry-over effect would be here. Carry over from which situation to which situation? Also, please justify why it is relevant to refer to an effect which is not statistically significant.

Response #6: We have now removed the reference to the “carry-over effect” of the non-significant effect. The paragraph is now more focused on the significant effects:

*** ‘Choice accuracy decreased across the acoustic distractor sessions (i.e., dogs tended to sensitise to this stimulus across the experiment) (session number: estimate = -1.821 (Std Error=0.523), z=-3.126, p<0.002). This decrease in accuracy was elevated in dogs that dropped out earlier (session number*drop-out stage: estimate =0.409 (Std Error =0.126), z=3.327, p=0.001). It is notable that no dogs exhibited enduring behavioural changes to light or acoustic stimuli, see the paragraph titled ‘Trial drop-out. sensitisation and dog welfare’.’

DISCUSSION

Comment #7: Please add a section on the limitations of the study. This should address the small sample size, the large inter-individual variation (according to the statistics reported e g on lines 355-356 and 377-378 this accounted for almost all the variance explained by the model) and the potential for inferring from the study to actual working dog challenges. Also address, either here or in the relevant part of the Results section, how you dealt in the statistical analysis with the high drop-out rates.

Reply #7: We have now addressed the sample size issue in the results by conducting a power analysis.

*** ‘A power analysis was conducted using the R package SIMR in to assess whether our sample size (N=12 dogs, with 48, 72 and 48 observations per dog during the strobe, dark adaptation and acoustic stimuli condition, respectively) was large enough to test detect statistically meaningful effects (Green & MacLeod, 2015). This power analysis suggested that a sample size of n ≥ 9 was required to detect an effect of our fixed factors on dogs’ task performance with 83% power at a .05 significance level, thus our sample size (N = 12 dogs) was sufficient.’ 

We also included a sentence on how we dealt with the statistical analyses with the high drop-out rates (this was initially included in the supplementals and has now been moved to the main text):

*** ‘Sessions for which dogs did not complete every trial (i.e., drop-out sessions) were not included in our analyses. ‘Session number’ did not impact task performance in any of our analyses. Therefore, to avoid biassing our dataset (by omitting data from the dogs that sensitised to the acoustic distractor), we used a ‘last observation carried forward’ technique to replace missing follow-up observations with previous observations. This technique was suitable to replace the missing data as there were no trends for dogs to change their performance across different sessions (see Results section). Our results did not differ when uncompleted sessions were omitted from the data set.’

As recommended by the editor, we have also discussed the identified limitations of our study in a section called ‘Study limitations’:

***’

Study limitations

While our sample size was relatively low, N =12 dogs, power analyses confirmed that it large enough to detect statistically meaningful effects. Inter-individual variation in task performance was high and explained the greatest proportion of variation in our task performance data. Nonetheless, the strobe and acoustic stimulus tended to affect individual performance in a consistent manner– such that significant, negative performance effects were detectable. Our study has helped to highlight how different distractors impair task performance in dogs. However, there are potential limitations for inferring our findings to the actual challenges working dogs may face. For example, working dogs are often exposed to many distractors of many different modalities, which may occur simultaneously. It is unclear how a combination of different stimuli may impact dogs’ task performance, and thus future work is necessary to identify the specific (or specific combination of) stimuli that dogs find most distracting, to properly inform effective desensitisation strategies. Additionally, our sample size was not large enough to explore how dog breed may have affected the impact of stimuli on task performance – for example, our data could not distinguish whether prick eared dogs (characteristic of police dogs) are more sensitive to acoustic distraction than lop eared dogs, which is certainly an avenue for future work.’

Comment #8: In the final paragraph of the Discussion, you reflect on the implications for how working dogs should be trained. This is a relevant observation, but it should be complemented with a reflection on how dogs are trained in professional settings presently. I would be surprised if this aspect is not taken into account at least to some extent in existing professional dog training programs. There may not be scholarly papers, but there must be other sources of information. A quick search turned up a one-hour webinar from the US https://www.youtube.com/watch?v=nfB2L_NldA8 on how to manage distractions in police dog work. It may be that the easily available information does not address distractions of the type you have tested for, but you need to provide some evidence of how you have searched for this information.

Response #8. We have now acknowledged that many working dog organizations conduct desensitization programs to habituate their dog to various distractors. In line with the editors comments, we have now added this paragraph:

***’ Our study suggests that habituating dogs to dazzling lights (i.e., dark adaptation) is unlikely to improve task performance, and that time and resources may be better spent desensitising dogs to strobe (or flickering) lights and acoustic stimuli. Indeed, the gradual introduction of loud noises to the training of improvised explosive device (IED)-detection dogs has been shown to improve the performance of improvised explosive devices (IED)-detection dogs in the field (Christensen et al., 2006, Jones and Gosling, 2005). Additionally, repeated exposure to visuospatial distractors has been shown to improve performance and information filtering mechanisms in humans (Kelley and Yantis, 2009). Thus, incorporating strobe lighting into canine training regimes could also help to mitigate the performance impairments associated with this stimulus (however, more work is needed to effectively inform such regimes). Notably, the implementation of such desensitisation programmes should be conducted after the initial stages of training (once criterion has been met), as exposure to distractors during the initial stages of training has been shown to hinder the rate of task learning in dogs (Sheldon et al., 2023).’

REFERENCES

Comment #9: Please carefully check the references in the text against the reference list. The order in the list should correspond to the order in which the references appear in the text. For example, he second reference in the text is listed as the third in the list. Please also check the formatting of the references. Journal titles are sometimes capitalized, sometimes not, even for the same journal. Conference references are incomplete (should include location and date). Please see sample references here https://www.nlm.nih.gov/bsd/uniform_requirements.html

Response #9: We apologise for these mistakes/typos. We have now more thoroughly revised our citations and reference list according to the requirements provided by the editor. 

Comment #10: When referring to a source which is not a peer-reviewed paper, it should be clear from the text what the source is. When you use a company reference to the specifics of a piece of equipment, this is evident. But you also use industry references as sources for information on dogs’ hearing threshold, e g line 211. Is this the best source for this information?

Response #10: We have removed the non-peer reviewed references on line 211. We have also now added the source of the non-peer reviewed article. 

Reviewer comments.

Reviewer #1

Comment 1: The study is interesting, but it is too preliminary and too small. The methods lack rigor and do not meet the high standards of this journal. It also lacks rigor. The extremely small sample and too basic info disallow me to accept the paper in this journal with high standards. I suggest the authors augment the data to for example 40, 50 dogs or more and then reanalyze the data and resubmit their paper to PLOS One.

Reply 1: We have addressed the sample size issue. We conducted power analyses which suggested that our study design and sample size of n=12 dogs were sufficient to detect statistically significant differences. 

*** ‘A power analysis was conducted using the R package SIMR in to assess whether our sample size (N=12 dogs, with 48, 72 and 48 observations per dog during the strobe, dark adaptation and acoustic stimuli condition, respectively) was large enough to test detect statistically meaningful effects (Green & MacLeod, 2015). This power analysis suggested that a sample size of n ≥ 9 was required to detect an effect of our fixed factors on dogs’ task performance with 83% power at a .05 significance level, thus our sample size (n = 12 dogs) was sufficient.’ 

Reviewer #2

Comment 1: This study is a valuable contribution to the literature and clearly has practical relevance for working dog researchers and handlers. The findings are logically presented, analysed, and discussed. One limitation of the work is that only 12 dogs participated, and a considerable amount of inter-individual variability would be expected for this effect. However, this is a common limitation for experiments involving time-intensive training and testing and fortunately this sample of dogs did demonstrate a range of possible responses. I am satisfied that the conclusions are informative and not overstated.

Reply 1: Thank you. In line with the comments by reviewer #1 and the editor, we have more formally confirmed the capacity for our sample size to detect statistically meaningful trends:

 *** ‘A power analysis was conducted using the R package SIMR in to assess whether our sample size (N=12 dogs, with 48, 72 and 48 observations per dog during the strobe, dark adaptation and acoustic stimuli condition, respectively) was large enough to test detect statistically meaningful effects (Green & MacLeod, 2015). This power analysis suggested that a sample size of n ≥ 9 was required to detect an effect of our fixed factors on dogs’ task performance with 83% power at a .05 significance level, thus our sample size (n = 12 dogs) was sufficient.’ 

Comment 2: I also commend the researchers for the welfare measure of allowing dogs to initiate their own participation.

Reply 2: Thank you, welfare was an important consideration of the study.

Comment 3: Minor comments:

I would be interested to know the actual breeds of participating dogs (rather than just “working dog types”). Individuals’ sensitivity to visual and acoustic stimuli may at least partly be determined by breed.

Reply 3: Thank you, the breeds of the participating dogs can now be found in the Supplementary materials:

***Supplemental Table 1. The name, breed, sex, sexual status, age, ear type and eye colour of the dogs who took part in the experiment. ‘GSD’ = German Shepherd, ‘NSDTR’ = Nova Scotia Duck Tolling Retriever, ‘x’ refers to a cross breed.

Name Breed Sex Sexual status Age (years) Ear type Eye colour

Alexis GSD Female Intact 4 Pricked Brown

Arlo Labrador x Male Intact 1 Folded Brown

Ava Alaskan Malamute Female Neutered 6 Pricked Gold

Sky Kelpie Female Neutered 4 Pricked Gold

Lyra Labrador x Female Intact 1.5 Folded Brown

Poppy GSD x Malinois Female Neutered 3.5 Pricked Brown

Hunter Siberian Husky Female Neutered 6 Pricked 1 Blue, 

1 Brown

Hector NSDTR Male Neutered 5 Folded Gold

Lilly Border Collie Female Intact 1.5 Rose Brown

Logan Border collie Male Intact 3 Pricked Brown

Max Welsh Springer Spaniel Male Intact 1 Lop Brown

Pan Siberian Husky Female Neutered 9 Pricked Brown

Comment 4: Under the heading “stimuli used” lines 190-199, it is not immediately clear which room each type of stimuli were presented in (e.g., I was unsure in this section what the lighting conditions were for the dark adaptation condition in the holding vs testing room). This is expanded upon later, line 236 onwards, but clarity may be improved by including this piece of information in the first instance.

Reply 4: This has now been edited, and the stimuli used section has been integrated into the testing procedure section to avoid repetition and improve clarity per the reviewers’ suggestions:

*** ‘Dark adaptation condition: Before each dark adaptation testing session, dogs were kept with a researcher in a dark room (the holding room, without windows or lights) for twenty minutes. After twenty minutes, trials began, and during the stimulus trial the holding room was flooded with an intensely bright light for one minute. The light was either white (RGB: 255,255,255) (2400 lux), red (255,0,0) (610 lux) or blue (0,0,255) (1130 lux) (Supplementary Figure 2). After one-minute of bright light exposure, dogs were led by an experimenter into the testing room (low-level lighting; 2.5 lux) and manoeuvred into the start position. Dogs were kept in the start position for three seconds before being released to attempt the task on the command ‘touch’. Dogs were kept in the dark with a researcher throughout the dark adaptation sessions. Each dog experienced six presentations of each colour stimulus, 18 sessions in total.

Strobe lighting condition: Before each trial, dogs were kept with a researcher in the holding room (lights turned on, ambient lighting approximately 350 lux) for three minutes, after which they were led into the testing room (2.5 lux), manoeuvred to the starting position for three seconds before being released to attempt the task on the command ‘touch’. During the stimulus trial, a strobe light (with an intensity of 100,000 lux and a flash rate of either 5, 10 or 15 Hz) was presented as soon as the dog proceeded to move down the ladder runway. For reference, 500 lux is equivalent to the brightness of a laptop screen, and 100,000 lux is equivalent to direct sunlight (The Engineering Toolbox, 2004). Dogs experienced two trials at each speed, creating 6 sessions in total. To reduce the risk of any cumulative effects from the stimulus on dogs an inter trial interval of one1 hour was used between each session. 

Acoustic distractor condition: Before each trial, dogs were kept at ambient lighting (350 lux) in the holding room for 1 minute before being led into the testing room (2.5 lux) and manoeuvred into the starting position. During the stimulus trial a 85dB acoustic distractor was played as soon as the dog started to walk over the ladder obstacle. To reduce the risk of any cumulative effects from the stimulus on dogs an inter session interval of at least 1 hour was used. Each dog experienced a maximum of six sessions; however, if they refused to start the ladder on three subsequent trials, or showed any reluctance to re-enter the testing room their participation was terminated, and the dog was considered to have ‘dropped-out’ of

---

## [Editor Report · Decision Letter 1]

22 Nov 2023

Working dogs in dynamic on-duty environments: the impact of dark adaptation, strobe lighting and acoustic distraction on task performance

PONE-D-23-15303R1

Dear Dr. Sheldon,

We’re pleased to inform you that your manuscript has been judged scientifically suitable for publication and will be formally accepted for publication once it meets all outstanding technical requirements.

Kind regards,

I Anna S Olsson, Ph.D.

Academic Editor

PLOS ONE
---

## [Editor Report · Acceptance letter]

30 Nov 2023

PONE-D-23-15303R1 

Working dogs in dynamic on-duty environments: the impact of dark adaptation, strobe lighting and acoustic distraction on task performance 

Dear Dr. Sheldon:

I'm pleased to inform you that your manuscript has been deemed suitable for publication in PLOS ONE. Congratulations! Your manuscript is now with our production department. 

Kind regards, 

on behalf of

Dr. I Anna S Olsson 

Academic Editor

PLOS ONE